# Spectral Analysis Methods for Improved Resolution and Sensitivity: Enhancing SPR and LSPR Optical Fiber Sensing

**DOI:** 10.3390/s23031666

**Published:** 2023-02-02

**Authors:** Paulo S. S. Dos Santos, João P. Mendes, Bernardo Dias, Jorge Pérez-Juste, José M. M. M. De Almeida, Isabel Pastoriza-Santos, Luis C. C. Coelho

**Affiliations:** 1INESC TEC—Institute for Systems and Computer Engineering, Technology and Science, Rua Dr. Alberto Frias, 4200-465 Porto, Portugal; 2FEUP, University of Porto, R. Dr. Roberto Frias, 4200-465 Porto, Portugal; 3FCUP, University of Porto, Rua do Campo Alegre, 4169-007 Porto, Portugal; 4CIQUP/IMS—Chemistry Research Unit, FCUP, University of Porto, 4169-007 Porto, Portugal; 5CINBIO, Universidade de Vigo, Campus Universitario Lagoas, Marcosende, 36310 Vigo, Spain; 6SERGAS-UVIGO, Galicia Sur Health Research Institute, 36312 Vigo, Spain; 7Department of Physics, University of Trás-os-Montes e Alto Douro, 5001-801 Vila Real, Portugal

**Keywords:** plasmonics, optical fiber sensors, plasmonic sensing, spectral analysis, SPR, LSPR

## Abstract

Biochemical–chemical sensing with plasmonic sensors is widely performed by tracking the responses of surface plasmonic resonance peaks to changes in the medium. Interestingly, consistent sensitivity and resolution improvements have been demonstrated for gold nanoparticles by analyzing other spectral features, such as spectral inflection points or peak curvatures. Nevertheless, such studies were only conducted on planar platforms and were restricted to gold nanoparticles. In this work, such methodologies are explored and expanded to plasmonic optical fibers. Thus, we study—experimentally and theoretically—the optical responses of optical fiber-doped gold or silver nanospheres and optical fibers coated with continuous gold or silver thin films. Both experimental and numerical results are analyzed with differentiation methods, using total variation regularization to effectively minimize noise amplification propagation. Consistent resolution improvements of up to 2.2× for both types of plasmonic fibers are found, demonstrating that deploying such analysis with any plasmonic optical fiber sensors can lead to sensing resolution improvements.

## 1. Introduction

Nanostructures capable of supporting plasmonic resonances have been studied extensively in the last decades, finding applications in imaging, photovoltaics, and sensing, among others [1,2,3,4,5,6,7,8,9]. Typically, the excitation of plasmonic resonance depends on the availability of free electrons in the conduction band of the nanomaterial, nanostructure geometry, nanoparticles (NPs), and inter-particle spacing [10,11,12,13]. Plasmonic resonances can be categorized into two groups: localized surface plasmonic resonance (LSPR) and propagating surface plasmonic resonance (SPR). The former is typically associated with well-dispersed NPs, and presents a strong EM-field around the NP surface, whereas the latter shows an EM-field capable of reaching hundreds of nanometers [14,15]. These two types of plasmonic resonances present very different optical properties, as the sensitivity to refractive index (RI) changes in the surrounding medium. In fact, SPR RI sensitivity is still about 4 orders of magnitude larger than the localized counterpart [16].

Regarding the excitation of such plasmonic resonances, NPs do not require special optical platforms as in the case of film-based SPR. To enable the use of film-based structures, specific platform configurations were developed from classic prism-based Kretschmann and Otto configurations to fiber-based platforms. The latter are of special interest since they offer a wide range of new possible configurations [17]. It is worth mentioning that LSPR with NP can also be excited on both planar and optical fiber platforms. A typical optical configuration to enable effective plasmonic sensing with optical fibers is shown in Figure 1. Here, a small decladded section is covered with a metallic thin film or with metallic NPs. This allows for the light traveling on the fiber core to interact with the nanostructure, exciting a plasmonic band. The use of multi-mode fibers involves an angular light distribution that will interact differently with the nanostructure. Then, the number of interactions with the plasmonic structure depends on the angle of incidence. This parameter is called the angular interaction distribution factor and is inversely proportional to the tangent of the angle [18]. Therefore, the obtained spectra can be represented as the sum of all individual angular contributions.

The combination of the optical platform and nanostructure results in complex spectral characteristics that respond in a unique way to external medium changes. This response involves symmetrical and asymmetrical band broadening, band amplitude variations, and band wavelength shifting. In sensing, the increase in such variations through careful physical nanostructure tailoring has been focused on in the research [12,19,20,21,22]. However, an often-overlooked aspect is the analysis method itself since sensitivity is usually only characterized by wavelength shifting or amplitude variations in the plasmonic band peak.

The question of monitoring a broader range of spectral changes was already partially addressed by other authors on works with gold NPs of different geometries on planar structures. To date, the method that reported the highest improvements in RI sensitivity is the inflection point (IF) method for gold NPs of different geometries, as presented by Chen et al. [23]. The method is based on the differentiation of the plasmonic band, tracking the first derivative local minima. Such a methodology is based on the unsymmetrical broadening of the plasmonic band for increasing RI, causing larger wavelength shifts on the inflection point compared to the plasmonic band peak. The authors reported RI sensitivity increments of between 18% and 55% for gold nanospheres (NS), nanostars, and nanorods. Other works also applied the same method on other gold nanostructures showing consistent improvements [24,25]. Chen et al. [26] developed another approach for small gold NS based on the plasmonic band’s second derivative minima, which can be related to curvature variations around the band peak. The authors reported increased signal-to-noise ratios, culminating in increased measurement resolutions. However, for larger NS, where the extinction cross section is not solely dominated by absorption, scattering becomes relevant, rendering such an approach inaccurate.

The NP complex permittivity, since it is based on spectral variations of the plasmonic band, will play a major role in such variations. As so, claiming that it can be applied to other nanomaterials (e.g., silver), without further investigation, is merely speculative. The reported works completely lack any reasoning for their applicability on thin film-based SPR bands. Finally, as the optical platform plays a major role in the resulting band, especially when working with optical fiber sensors, further investigation is needed to assess the viability of the method when working with optical fibers. Finally, the explanation behind the research focus on NPs can be attributed to the NP’s intrinsic low RI sensitivity when compared to thin film plasmonics. As so, the study on the analysis methods capable of resolution enhancement for SPR sensors still needed to be explored. Herein, a general approach based on differentiation methods for the study of spectral variations on both SPR and LSPR sensors will be presented both numerically and experimentally. Differentiation will be made using total variation regularization differentiation, in order to mitigate noise amplification problems. The study will be conducted on a broader range of spectral features of the plasmonic band and its first two derivatives with gold or silver NS, as well as with gold or silver thin films on a multi-mode fiber-sensing structure. Then, the best methodology for each sensing structure will be individually assessed.

## 2. Materials and Methods

### 2.1. Chemicals

Gold salt (HAuCl${}_{4}\cdot$3H${}_{2}$O), silver nitrate (AgNO${}_{3}$), trisodium citrate (Na${}_{3}$C${}_{6}$H${}_{5}$O${}_{7}$), tannic acid (C${}_{76}$H${}_{52}$O${}_{46}$), polyethylenimine (H(NHCH${}_{2}$CH${}_{2}$)${}_{n}$NH${}_{2}$), and poly-sodium 4-styrenesulfonate ((C${}_{8}$H${}_{7}$NaO${}_{3}$S)${}_{n}$) were purchased from Sigma-Aldrich. All chemicals were used without any further purification. Milli-Q water (ρ = 18.2 MΩ) was used in all experiments. All glassware was successively cleaned with acetone and ethanol, rinsed with Milli-Q water, and stored at 60 ${}^{{}^{\circ}}$C before use.

### 2.2. Optical Fiber Functionalization with Au or Ag Nanoparticles

First, citrate-stabilized Au and Ag NS of 50 nm in diameter were synthesized by a kinetically controlled seeded growth method, as described by Bastus et al. [27,28]. After the preparation, the NS were centrifuged at 5000 rpm and redispersed in the same volume with Milli-Q water.

On the other hand, the fiber was functionalized with polyelectrolytes in a layer-by-layer assembly method. This process started by immersing the optical fibers in acetone for 1 h. Then, they were washed with soap and water, followed by an ultrasound bath for a few minutes. Finally, the fiber was washed with acetone and Milli-Q water. After the cleaning process, the optical fiber was immersed in an aqueous solution of PEI (poly(ethylenimine) 50% (*w/v*) in water) with a concentration of 1 mg/mL, containing 0.5 M NaCl for 30 min. Then, the fiber was washed three times by immersion in Milli-Q water for 1 min. Next, the optical fiber was immersed in an aqueous solution of PSS (poly-sodium 4-styrenesulfonate) with a concentration of 1 mg/mL, containing 0.5 M NaCl, for 20 min. Again, the fiber was washed three times by immersion in Milli-Q water for 1 min. The process was finalized by assembling another PEI layer following the same procedure for 20 min, and the same final washing with Milli-Q water.

### 2.3. Optical Fiber Functionalization with Au or Ag Thin Film

The metal films were deposited on the optical fiber structure using radio-frequency (RF) magnetron sputtering. Thus, the fibers were placed inside the chamber in a rotating system, which allowed a uniform deposition around the exposed core. Prior to the metal deposition, a 3 nm Cr layer was first deposited in order to ensure better adhesion. For both gold and silver, the deposition process was done at a base pressure of 8 × 10${}^{-3}$ mbar and a power of 20 W, ensuring a high deposition rate of 0.15 nm/s, as recommended for plasmonic applications [29]. The thickness was controlled by a film thickness monitor (Edwards, Ltd., FTM5, West Sussex, UK, resolution of 0.1 nm) along with a previously made calibration. The final metal thickness was 50 nm.

### 2.4. Instrumentation

The developed sensing structure is based on a silica core multi-mode fiber (MMF-FT600UMT, Thorlabs, Dortmund, Germany) with a core/cladding diameter of 600/625 μm, respectively, and a 1 cm long sensing section. The fiber core was exposed by the mechanical removal of the plastic fiber jacket and the hard fluoropolymer cladding. Then, both ends were polished using 8 and 3 μm polishing disks (Fibermet, Buehler, Lake Bluff, IL, USA) until a flat and optical quality surface was reached. Finally, the fiber was washed in a mixture of detergent with pure water in an ultrasonic bath for 10 min.

The optical response was recorded by connecting the fiber section through SMA905 connectors and multi-mode fiber patch cords to a tungsten halogen light source (Ocean Optics, HL-2000, Dunedin, FL, USA) and to a high-resolution spectrometer (Speed+, SarSpec, Porto, Portugal). The absorption spectra were acquired on a wavelength range from 200 to 1050 nm, with a resolution of 1.7 nm, and an integration time of 10 ms. The sensitivity to the surrounding RI was determined by immersing the fiber sensing section in mixtures of glycerol and Milli-Q water at different volume ratios, presenting RI values from 1.333 to 1.415.

### 2.5. Simulation of Plasmonic Nanostructures

The simulation setups for both thin-film and nanoparticle-based optical fiber plasmonic sensors are shown in Figure A1, along with the light angular distribution, with data measured by Yasukawa et al. [30]. The interaction correction factor is given by N = L/(D tan(θ)), where L is the sensor length and D is the fiber diameter [18]. The simulation output was then obtained by integrating all of the individual angles according to their interaction factors. The angular dependencies on the final spectra can be seen in Figure [28].

The simulation of the SPR sensor on the multi-mode optical fiber platform was done using a transfer matrix method [31]. The fiber core was modeled as an infinite layer of glass, covered by a 50 nm gold or silver thin film, with another infinite dielectric layer on top, representing the external medium, using both s and p polarizations with equal magnitudes to mimetize unpolarized light behavior. The simulation of the LSPR sensor was based on a boundary element method [32]. As a model, one gold or silver NS of 50 nm, deposited over the infinite glass layer, was simulated under both polarizations. The material permittivity was obtained through the python package PyTMM [33]. For gold and silver, data were taken from Johnson and Christy [34]. For glass, data were taken from Malitson [35].

The thicknesses (50 nm) of both the gold and silver thin films were chosen due to the optimization of the resulting plasmonic band properties, namely in terms of RI sensitivity over the band’s full width at half-maximum [36]. The NP diameter of 50 nm was chosen to match thin film thicknesses while presenting optimization in terms of the LSPR band’s intensity and width [28].

### 2.6. Data Analysis

The spectral features herein studied are the wavelength and amplitude variations of the band peak, as well as the minima of the first and second derivatives. This is illustrated in Figure 2. In order for this work to be coherent with the literature, the band peak tracking will be referred to as (S), the first derivative as (IF), and the second derivative as (K). Moreover, to distinguish between amplitude variations and wavelength shifts of such points, wavelength shift acronyms will be followed by (λ) and amplitude variations by (A).

The study of another set of parameters that requires performing mathematical operations, such as sums or multiplications on such spectral features, are not considered, since the error propagation analysis will lead to larger errors. This problem does not arise when performing numerical differentiation. As so, it will not lead to an error increase. However, with real measurements, multiple noise sources from the instrumentation setup to the optical sensor itself will leak into the acquired spectra. When performing differentiation, all noise in the signal will propagate and be amplified throughout the analysis. This can lead to an unreadable signal if special care with noise is not taken. At first glance, one can try to apply some type of low-pass filter to denoise the signal before and/or after performing differentiation, but this will lead to poor results with significant distortion. Nevertheless, with regularized differentiation, this problem can be tackled effectively [37]. More details can be found in Appendix B. The difference between standard and regularized differentiation at different levels can be directly compared in Figure A3. In this work, the regularized differentiation was performed directly over the raw data, using the open-source Python library Pynumdiff [38]. To calculate the standard deviation for each sensor, a linear fit of each spectral feature maxima/minima was performed and the errors were assessed. Resolution (R) was then calculated by dividing this standard deviation (σ) by the RI sensitivity (S) as R = σ/S.

## 3. Results

### 3.1. Simulation of Plasmonic Nanostructures on Optical Fiber

The performances of two types of plasmonic optical fiber sensors were simulated employing the transfer matrix method (see the experimental section for further details). As models, we employed a multi-mode optical fiber coated with a gold or silver NS, with a diameter of 50 nm, as represented in Figure 3A,B, and a multi-mode optical fiber coated with 50 nm gold or silver thin film. For the data analysis, the spectral features herein studied are the wavelength and amplitude shifts of the LSPR/SPR band, as well as the minima of the first and second derivatives, as shown in Figure 2. The simulated responses of all those structures to external RI variations are displayed in Figure 3.

As shown in Figure 3A, the gold NS structure exhibits the highest wavelength shift when employing the IF method with a RI sensitivity of 112 nm/RIU. This nearly represents sensitivity doubling when compared to the S and K wavelength variations. Amplitude-wise, the IF method also showed the highest variation, although with a marginal improvement when compared to the traditional S-peak tracking. Regarding the fiber structure coated with silver NS (Figure 3B), among the wavelength-tracking methods, the IF method only presented a marginal increment over the S and K features. Regarding amplitude variations, S showed the highest sensitivity, whereas K presented the lowest sensitivity. Such results indicate that the smaller band-broadening obtained with silver-based plasmonics, in comparison to the gold-based one, is responsible for the smaller effectiveness of the IF method. Nevertheless, the IF method was observed to present the highest sensitivity in all cases.

The numerical responses of both gold or silver thin films on the optical fiber are shown in Figure 3C and Figure 3D, respectively. Regardless of the nature of the metal, the best RI sensitivity was obtained for the IF method, both in wavelength and amplitude tracking methodologies. Namely, a sensitivity increase from 3100 to 3600 nm/RIU in wavelength tracking from S to IF, and a relative amplitude variation from 5 to 13 RIU${}^{-1}$, for the gold film. For the silver film, a sensitivity increase from 4050 to 4600 nm/RIU from S to IF, and a relative amplitude variation from 8 to 13 RIU${}^{-1}$, were obtained. These results successfully indicate the possibility of improved measurements by tracking other spectral features, based on the differentiation of the main spectra. Additionally, the previously unreported IF-amplitude feature is here shown to be capable of performing better than the other two amplitude tracking features, for both SPR sensors (gold and silver).

### 3.2. Experimental Characterization of Plasmonic Optical Fibers

The plasmonic optical fiber sensors based on NPs were fabricated via the deposition of 47 nm Au or 52 nm Ag NS (see Figure A4) on the multi-mode optical fiber following a layer-by-layer approach through electrostatic interactions. The nanoparticles are uniformly distributed on the fiber as shown in Figure A5. Next, the RI sensitivity was investigated by recording the optical response when immersing within different mixtures of glycerol–water.

The results showed that gold NS presented similar RI sensitivities for the three wavelength tracking features analyzed (see Figure A6), namely, around 650 nm/RIU. This experimental RI sensitivity was much larger than the simulation data (IF(λ) method with a RI sensitivity of 112 nm/RIU). This RI sensitivity increment can be attributed to inter-particle coupling and the creation of hotspots as reported in [39]. Nevertheless, the experimental IF(λ) method showed the highest and K(λ) the lowest sensitivity; thus, in agreement with the simulations. With regard to amplitude variations, the K method presented the largest variations, as well as higher standard deviations; thus, not rendering this method as ’not suitable’ to achieve resolution enhancements. The overall best resolution was obtained with the IF(λ) method, showing a resolution of 1.74 × 10${}^{-3}$ RIU${}^{-1}$. This represents a 2.2× resolution increment when compared to traditional S(λ) tracking.

The optical fiber sensor functionalized with silver NS also presented an overall larger sensitivity in terms of wavelength tracking when compared to the simulations. This can be attributed to inter-particle coupling and the creation of hotspots. Furthermore, the smaller sensitivity (542 nm/RIU), when compared to the studied gold NS, can be understood by a smaller density of NS over the fiber surface (see Figure A5B). Among wavelength tracking-based methods, the IF presented the highest sensitivity of them all, which is in agreement with the numerical results, achieving the best resolution among the wavelength tracking techniques, with a resolution improvement of 1.2×, when compared to traditional S(λ) tracking. However, in this case, the best resolution came from the S(A) tracking, presenting a 2.1× resolution increment when compared to its S(λ) counterpart. Thus, showing that amplitude tracking, in some cases, can surpass wavelength tracking. As so, amplitude-based tracking should not be promptly discarded for such plasmonic sensing structures.

With respect to the 50 nm thin film optical fibers, their RI sensitivity is higher than their LSPR counterparts. As shown in Figure 4C,D, they exhibit RI sensitivities ranging from 2000 to 3300 nm/RIU for the gold and silver films, respectively. These values are smaller than the simulations (3600 and 4600 nm/RIU for the gold and silver thin films, respectively). This can be attributed to numerous factors, such as surface roughness and film thickness variability, among others. Even though S, IF, and K analyses show similar behaviors (see Figure A6), the best resolution was again achieved using the IF(λ) analysis, showing values of 0.98 × 10${}^{-3}$ RIU${}^{-1}$ and 1.28 × 10${}^{-3}$ RIU${}^{-1}$ for the gold and silver films, respectively.

As the sensing characteristics are highly dependent on the sensor construction characteristics (e.g., the fiber surface coverage with NP), a complete study of all three points in both amplitude and wavelength variations should be done, in order to assess the best methodology for each specific case.

### 3.3. Reproducibility of Gold Nanoparticle-Based Optical Fiber

To assess the practicability of the use of the three methods of analysis, four different gold nanoparticle-based optical fibers were investigated (see the SEM analysis and refractometric responses in Figure A7 and Figure 5).

Despite some surface NP density variations, a general trend in the measured sensitivity was obtained for all samples. Namely, IF(λ) shows the highest average sensitivity of 690 nm/RIU, followed by K(λ) and S(λ) with 613 nm/RIU and 585 nm/RIU, respectively. Although the K(λ) method presented, on average, a higher sensitivity than S(λ), the noticeable dispersion increase can hinder the benefits of its usage. Amplitude-wise, the IF(A) presented the greatest RI sensitivity increase when compared to S(A), from 0.65 to 0.89 A/RIU.

Regarding resolution, the IF(λ) method presented the best results with a resolution of 3.0 ± 1.1 × 10${}^{-3}$ RIU${}^{-1}$, an increase of 62% when compared to the traditional S(λ) method (4.9 ± 1.4 × 10${}^{-3}$ RIU${}^{-1}$). Thus, showing that it can effectively be a useful analysis technique to increase measurement resolution. On the other hand, the amplitude-tracking methods, in general, performed worst. This reproducibility test demonstrated that a similar trend was obtained for four equal plasmonic optical fiber sensor constructions.

## 4. Conclusions

In this work, it was found that the sensor fabrication variability, such as NP density over the fiber, can affect the sensing performance. As such, no specific spectral feature can be categorized as the best-performing. Nevertheless, by tracking and comparing the peaks of the absorbance spectra and the first two derivatives, it is possible to arrive at an optimum spectral feature to monitor. In the case of Au NP, it was found that a measurement resolution increase of 62% was found by using the IF(λ) method, which showed a resolution of 3.0 ± 1.1 × 10${}^{-3}$ RIU${}^{-1}$. It was also found that the same IF(λ) performed best for both thin film-based optical fiber sensors with resolutions of 0.9 × 10${}^{-3}$ RIU${}^{-1}$ and 1.3 × 10${}^{-3}$ RIU${}^{-1}$ for the gold and silver thin film-based sensors, respectively. The exception to such a trend came from the silver NP sensor, where the S(A) presented the best performance, showing a resolution of 1.0 ± 1.1 × 10${}^{-3}$ RIU${}^{-1}$.

To successfully manage the differentiation procedure, care must be taken with smoothing and low-pass filters, as they can drastically reduce the signal-to-noise ratios of both derivatives. Here, it was shown that total variation regularization differentiation suits such applications with high effectiveness. Since the presented methods are based on differentiation, they can readily be applied to measurements with plasmonic-based sensors, regardless of the nanomaterial or type (LSPR or SPR), potentiating their performances.

To the best of our knowledge, the direct nanomaterial comparison (gold and silver) of optical fiber plasmonic sensing, was explicitly compared here for the first time with such differentiation methodologies. As such, this work could lay a broad basis for further applications of such methods on other plasmonic optical fiber sensing structures. Furthermore, when working with planar platforms, similar benefits could also be present, which could further increase this study’s impact in the future.

## Figures and Tables

**Figure 1 sensors-23-01666-f001:**
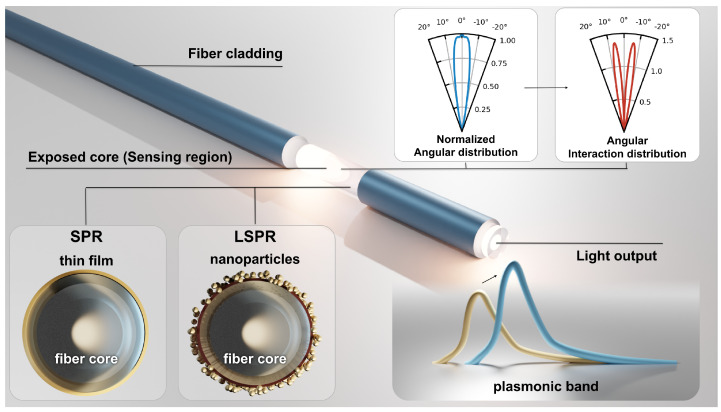
Multi-mode fiber with a decladded section exposing the core light to a plasmonic nanostructure; a thin film or nanoparticles deposited on its surface, exciting a SPR or LSPR band, correspondingly. Moreover, in the case of multi-mode fiber, there is a certain angular distribution and an interaction factor proportional to the traveling light angle.

**Figure 2 sensors-23-01666-f002:**
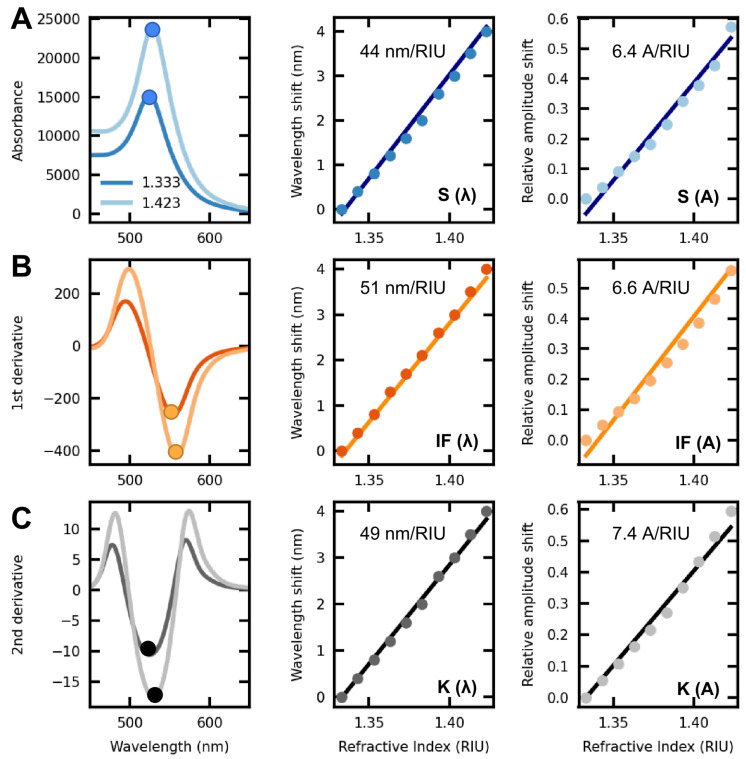
Scheme of the common optical response of a plasmonic nanostructure when subjected to external RI variations; (**A**) LSPR band (left) along S(λ) and S(A) shifts; (**B**) First derivative (left) along IF(λ) and IF(A) shifts; (**C**) second derivative (left) along K(λ) and K(A) shifts.

**Figure 3 sensors-23-01666-f003:**
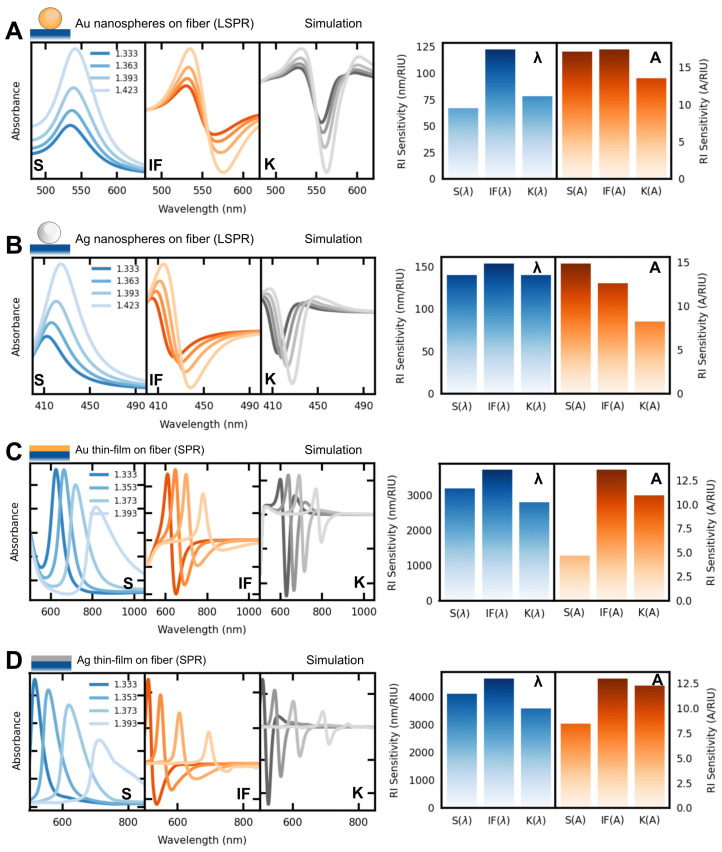
Simulated refractometric responses of different plasmonic optical fibers. Left panel: extinction spectra (S) as a function of the surrounding refractive index and the first two derivatives (IF) and (K), respectively. Right panel: RI sensitivity calculated from S, IF, and K; (**A**) gold NS, (**B**) silver NS, (**C**) gold thin film, and (**D**) silver thin film, on a multi-mode optical fiber (as schematically represented above the graphs).

**Figure 4 sensors-23-01666-f004:**
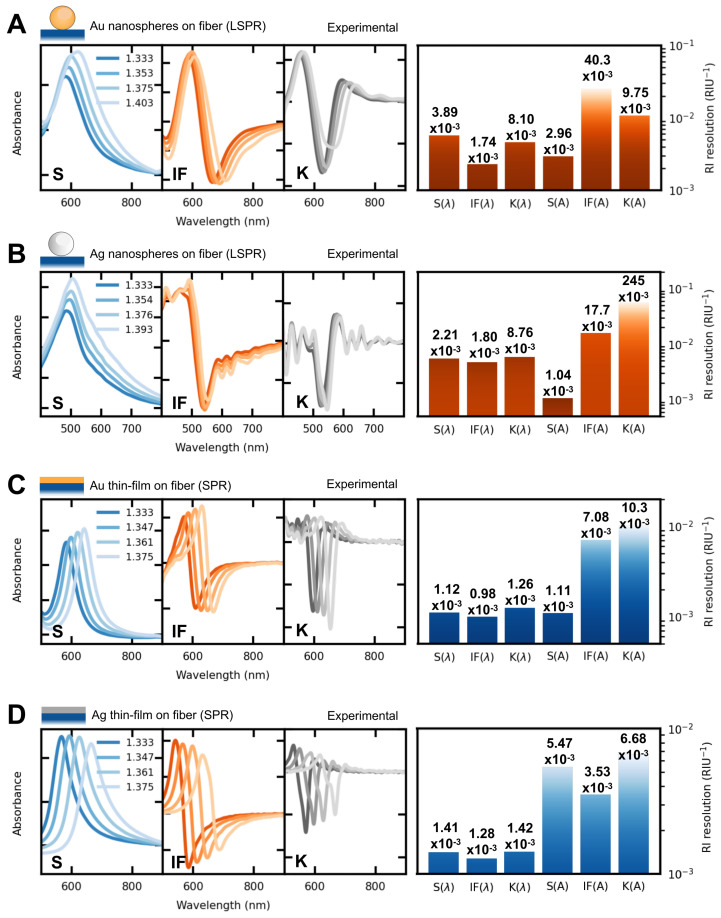
Refractometric response in terms of spectral changes and measurement resolution of (**A**) gold NS; (**B**) silver NS; (**C**) gold thin film; (**D**) silver thin film. The main spectra (and the first to the derivatives) were studied. In this way, the resolution for each method could be assessed.

**Figure 5 sensors-23-01666-f005:**
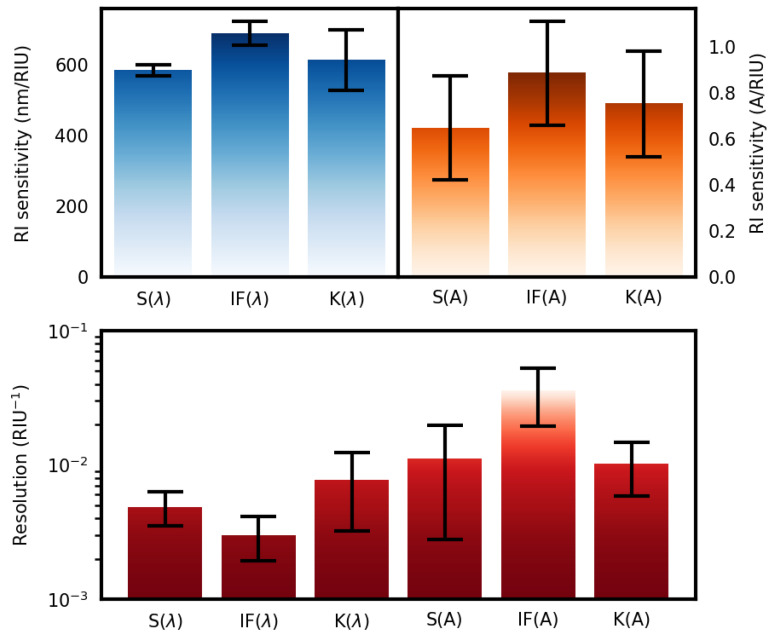
Top panel: Box plots of the RI sensitivity for the six analysis methods studied. Bottom panel: RI resolution comparison for the studied analysis methods.

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
