# Peer review of "Spectral Analysis Methods for Improved Resolution and Sensitivity: Enhancing SPR and LSPR Optical Fiber Sensing"

_sensors, 2023, doi:10.3390/s23031666_

Round 1
Reviewer 1 Report
Biochemical and chemical sensing with plasmonic sensors is widely performed by tracking the response of surface plasmonic resonance peaks to changes in the medium. In this work, such methodologies are explored and expanded to plasmonic optical fibers. The optical response of optical fibers doped gold or silver nanospheres and optical fibers coated with a continuous gold or silver thin film are studied experimentally and theoretically. Some issues should be clarified before considering to accept it for publication in Sensors.
1. It is said in the part of 2.5. Simulation of plasmonic nanostructures that the fiber core was modeled as an infinite layer of glass, covered by a 50 nm gold or silver thin film. Why chose the thick of the thin film to be 50nm? If the thick of the thin film is chosen to be thicker or thinner than 50nm, what phenomenon would be happened?
2. In the part of 3.2. Experimental characterization of plasmonic optical fibers, it is said that the plasmonic optical fibers based on NPs were fabricated via deposition of 47 nm Au or 52 nm Ag NS (see Figure A3) on the multimode optical fiber following a layer by layer approach through electrostatic interactions. How to control and measure the thick of deposition of Au is 47nm and Ag is 52nm?
3. In the part of 3.2. Experimental characterization of plasmonic optical fibers and 3.3. Reproducibility of gold nanoparticle-based optical fiber, there are “see Figure A3”, “in Figure A4”, “see Figure A5” and “FigureA6”, where are Figure A3, Figure A4, Figure A5 and FigureA6 in the manuscript?
4. It is based on the method of tracking the response of surface plasmonic resonance peaks to changes of RI to realize IR sensing. Why is it needed to obtain the 1st derivative and 2nd derivative of the absorbance curve in Figure 2, Figure 3 and Figure 4?
Author Response
We greatly appreciate the reviewer valuable comments and suggestions. According to the comments, we have revised the manuscript carefully. The answers to the questions raised are given as follows.
- It is said in the part of 2.5. Simulation of plasmonic nanostructures that the fiber core was modeled as an infinite layer of glass, covered by a 50 nm gold or silver thin film. Why chose the thick of the thin film to be 50nm? If the thick of the thin film is chosen to be thicker or thinner than 50nm, what phenomenon would be happened?
Our response: The thickness of 50 nm for both the gold and silver thin-films were chosen due to the optimization of the resulting plasmonic band properties, namely in terms of RI Sensitivity over the band Full-Width at Half-Maximum (FWHM). There are several works that explore explicitly the metal thickness dependency, such as: 10.1109/JSEN.2021.3080290 and 10.1016/j.snb.2016.05.117. As so, if one constructs a plasmonic sensor with smaller thicknesses, as the metal film gets smaller, the band intensity and overall sensing capacity will be greatly diminished. On the other hand, as one increases the metal thickness, it will reach a point where there will be not enough light intensity capable of reaching to the metal/external medium interface.
- In the part of 3.2. Experimental characterization of plasmonic optical fibers, it is said that the plasmonic optical fibers based on NPs were fabricated via deposition of 47 nm Au or 52 nm Ag NS (see Figure A3) on the multimode optical fiber following a layer by layer approach through electrostatic interactions. How to control and measure the thick of deposition of Au is 47nm and Ag is 52nm?
Our response: Effectively the sentence was not well constructed. As so, the sentence "The plasmonic optical fibers based on NPs were fabricated via deposition of 47 nm Au or 52 nm Ag NS (see Figure A3) on the multimode optical fiber following a layer-by-layer approach through electrostatic interactions. " was changed to "The plasmonic optical fiber sensors based on NPs were fabricated via deposition of 47 nm Au or 52 nm Ag NS (see Figure A3) on the multimode optical fiber following a layer-by-layer approach through electrostatic interactions"
The nanoparticle dimensions were controlled during their synthesis via kinetically controlled seeded growth method, as described by Bastus et al., as referenced in [26,27].
- In the part of 3.2. Experimental characterization of plasmonic optical fibers and 3.3. Reproducibility of gold nanoparticle-based optical fiber, there are “see Figure A3”, “in Figure A4”, “see Figure A5” and “FigureA6”, where are Figure A3, Figure A4, Figure A5 and FigureA6 in the manuscript?
Our response: When state Figure Ax, The A stands for figures in the appendix section. Please see the Appendix A through C to find such figures. This notation is in agreement with the sensors template.
- It is based on the method of tracking the response of surface plasmonic resonance peaks to changes of RI to realize IR sensing. Why is it needed to obtain the 1st derivative and 2nd derivative of the absorbance curve in Figure 2, Figure 3 and Figure 4?
Our response: This work explores the possibility of tracking other spectral features than the plasmonic band peak both in wavelength and in amplitude variations. Differentiation methods has previously shown potential to increase measurement sensitivity and resolution (please see references [22-25]).
Reviewer 2 Report
In this manuscript, the authors investigate surface-plasmon-based sensing with specially designed optical fibers. They study the optical response of optical fibers doped Au/Ag nanospheres and optical fibers coated with a continuous Au/Ag thin film. In sensing applications, resolution improvements of up to 2.2× for both types of plasmonic fibers are found. The results are solid and of practical value. The manuscript can be recommended for publishing in Sensors. There are some points for the authors to consider.
1. The authors use the transfer matrix method (TMM) for the simulation of their SPR sensors. If they can give more details about the calculation procedures, that will help the readers obtain more explicit understanding. By the way, they indicate the abbreviation “TMM” but do not use that in the following text.
2. In Figure A2, the insert legend covers too large part of the figure including data. I think the authors can try to avoid that or make the background color of the insert legend more transparent.
3. In Figure A5, it seems that the data points do not match the curves well, especially for the amplitude variation. Can the authors give more explanations?
Author Response
We greatly appreciate the reviewer valuable comments and suggestions. According to the comments, we have revised the manuscript carefully. The answers to the questions raised are given as follows.
- The authors use the transfer matrix method (TMM) for the simulation of their SPR sensors. If they can give more details about the calculation procedures, that will help the readers obtain more explicit understanding. By the way, they indicate the abbreviation “TMM” but do not use that in the following text.
Our response: The transfer matrix method has been widely used to simulate SPR sensors, as it provides faster simulations when compared to other mainstream simulation methods such as FEM or FDTD. Regarding the specificity of using such method to describe the multimode optical fiber response, details were described on section "2.5. Simulation of plasmonic nanostructures". To increase clarity, a figure describing the layers stack and light angular distribution was added to the appendix section. Please see Figure A1.
Thank you for noticing that the "TMM" abbreviation was not used throughout the text, as so it was removed.
- In Figure A2, the insert legend covers too large part of the figure including data. I think the authors can try to avoid that or make the background color of the insert legend more transparent.
Our response: Thank you for helping improve the figure. Please see the new changes to the figure (now Figure A3). The authors moved the legend to outside the figure and removed the duplicated legend on the right panel.
- In Figure A5, it seems that the data points do not match the curves well, especially for the amplitude variation. Can the authors give more explanations?
Our response: Thank you for noticing the discrepancy between the data points and fitting curves. This was just an error in the code to produce the figures, where, erroneously, it was normalized to the first point. Looking at the first point in all the plots they are all share the same coordinate, which obviously cannot be correct. We have now corrected this plotting error.
Although the differentiation with total variation regularization effectively mitigated largely noise amplification, it did not eliminate it completely. As so, the curves on the amplitude variations side, especially for the higher derivatives, were more affected due to their intrinsic higher standard deviations. The revised figure can be seen on Figure A6.
Reviewer 3 Report
The authors experimentally and theoretically investigated the optical response of optical fibers doped gold or silver nanospheres and optical fibers coated with a continuous gold or silver thin film. The study meets the scope of Sensors journal and the simulation and experiment results seem acceptable. Nevertheless, there are some minor issues that should be addressed:
1. In the introduction section, please elucidate the novelty and advantage of the proposed SPR optical fiber sensor compared to the SPR photonic crystal fiber sensor (e.g., Photonics 2022, 9(12), 916).
2. To enrich the background of the plasmonic effect of metal NS, a related reference (Optics Communications, 2016, 370, 85-90) is suggested to be included in the introduction section.
3. Why chose the size of 50 nm for Ag/Au NPs and thin films? The effect of the different size of Ag/Au NPs and thin films on optical response should be briefly describe in the text.
4. What kind of optical fiber used in the experiment should be clarified in more detail.
5. Line 173, what is the definition of σ.
6. The simulation setting should be clarified in more detail. The schematic diagram of simulation model is suggested to be given in the text.
7. Typo, e.g. line 237, “their rRI sensitivity is higher”. In the same manner, please check throughout the manuscript.
Author Response
We greatly appreciate the reviewer valuable comments and suggestions. According to the comments, we have revised the manuscript carefully. The answers to the questions raised are given as follows.
- In the introduction section, please elucidate the novelty and advantage of the proposed SPR optical fiber sensor compared to the SPR photonic crystal fiber sensor (e.g., Photonics 2022, 9(12), 916).
Our response: This work explores the possibility of tracking other spectral features than the plasmonic band peak in both wavelength and amplitude variations. Differentiation methods has previously shown potential to increase measurement sensitivity and resolution (please see references [22-25]). As so, we are not here developing an optical fiber plasmonic sensor structure as the work presented in (Photonics 2022, 9(12), 916). Furthermore, the work here presented could even be used to increase the measured resolution and sensitivity of (Photonics 2022, 9(12), 916).
- To enrich the background of the plasmonic effect of metal NS, a related reference (Optics Communications, 2016, 370, 85-90) is suggested to be included in the introduction section.
Our response: Thank you for the suggestion. Solar cell applications are indeed an interesting field where nanoparticle-based plasmonics are gaining attention. As so, the provided reference was added to the manuscript in line 19, reference 9.
- Why chose the size of 50 nm for Ag/Au NPs and thin films? The effect of the different size of Ag/Au NPs and thin films on optical response should be briefly describe in the text.
Our response: The thickness of 50 nm for both the gold and silver thin-films were chosen due to the optimization of the resulting plasmonic band properties, namely in terms of RI Sensitivity over the band Full-width at half-maximum. There are several works that explore explicitly the metal thickness dependency, such as: 10.1109/JSEN.2021.3080290 and 10.1016/j.snb.2016.05.117. For the NP, a diameter of 50 nm was chosen for three reasons: Nanospheres of such dimensions present a good compromise between LSPR band intensity and width; Diameters around 50 nm are typical dimensions, that can be found throughout the literature; Since we are comparing to 50 nm thin-films, the same size seemed a good value to work.
In section "2.5 Simulation of plasmonic nanostructures" the following sentences were added at the end: "The thickness of 50 nm for both the gold and silver thin-films were chosen due to the optimization of the resulting plasmonic band properties, namely in terms of RI Sensitivity over the band Full-width at half-maximum [36]. The NP diameter of 50 nm was chosen to match thin films thickness while presenting an optimization in terms of LSPR band intensity and width [28]"
- What kind of optical fiber used in the experiment should be clarified in more detail.
Our response: The optical fiber used in the experiment and the way it is modified is described in detail on section "2.4. Instrumentation".
- Line 173 what is the definition of σ.
Our response: σ represents standard deviation. On line 178 the sentence was changed to "Resolution (R) was then calculated by dividing this standard deviation(σ) by the RI sensitivity (S) as R = σ/S."
- The simulation setting should be clarified in more detail. The schematic diagram of simulation model is suggested to be given in the text.
Our response: Thank you for the suggestion, indeed on section "2.5. Simulation of plasmonic nanostructures " a description on the design of the simulation layers stack and nanoparticles is given. Nevertheless, to increase clarity, a figure describing the layers stack and light angular distribution was added to the appendix section. Please see Figure A1.
- Typo, e.g. line 237, “their rRI sensitivity is higher”. In the same manner, please check throughout the manuscript.
Our response: The typo on line 237 was corrected. Also was found a missing parenthesis in line 136 "N = L / (D tan(θ)", it was corrected to ": N = L / (D tan(θ))".
Round 2
Reviewer 1 Report
The authors have revised the manuscript carefully and it could be accepted for publication in Sensors.